The physiologic and therapeutic role of heparin in implantation and placentation

Quaranta Michela 1
Erez Offer 2 erezof@bgu.ac.il
Mastrolia Salvatore Andrea 3
Koifman Arie 2
Leron Elad 2
Eshkoli Tamar 2
Mazor Moshe 2
Holcberg Gershon 2
1 Department of Obstetrics and Gynecology, Azienda Ospedaliera Universitaria Integrata, Università degli Studi di Verona , Verona , Italy
2 Department of Obstetrics and Gynecology, Soroka University Medical Center, School of Medicine, Ben Gurion University of the Negev , Beer Sheva , Israel
3 Department of Obstetrics and Gynecology, Azienda Ospedaliera-Universitaria Policlinico di Bari, School of Medicine, University of Bari “Aldo Moro” , Bari , Italy
Remuzzi Giuseppe
Electronic publication date: 2015 Jan 6
Publication date: 2015
Volume: 3
Electronic Location ID: e691
Received 2014 Sep 3; Accepted 2014 Nov 19
Copyright: © 2015 Quaranta et al.
Copyright year: 2015
Copyright holder: Quaranta et al.
License: This is an open access article distributed under the terms of the Creative Commons Attribution License, which permits unrestricted use, distribution, reproduction and adaptation in any medium and for any purpose provided that it is properly attributed. For attribution, the original author(s), title, publication source (PeerJ) and either DOI or URL of the article must be cited.
License URL: https://creativecommons.org/licenses/by/4.0/

Keywords: Glycosaminoglycan, Trophoblast, Selectins, Cadherins, HB-EGF, Matrix metalloproteinases, Immune system

Funding: The authors declare that there was no funding for this work.

==============================
Implantation, trophoblast development and placentation are crucial processes in the establishment and development of normal pregnancy. Abnormalities of these processes can lead to pregnancy complications known as the great obstetrical syndromes: preeclampsia, intrauterine growth restriction, fetal demise, premature prelabor rupture of membranes, preterm labor, and recurrent pregnancy loss. There is mounting evidence regarding the physiological and therapeutic role of heparins in the establishment of normal gestation and as a modality for treatment and prevention of pregnancy complications. In this review, we will summarize the properties and the physiological contributions of heparins to the success of implantation, placentation and normal pregnancy.

Introduction

The use of heparins has increased since their discovery due to the number of properties and effects shared by these molecules. In addition to their anticoagulant and anti-inflammatory effects that justify their employment in the prevention and treatment of pregnancy complications, these molecules have a physiologic role during gestation and especially during implantation and placentation; this is critical in the establishment and success of pregnancy (Romero et al., 2002). In this review we will present the role of heparins in implantation, placentation and in the immunologic balance between the mother and her fetus, as well as discuss the clinical application of these concepts in the improvement of pregnancy outcome in patients with repeated implantation failure (RIF) and recurrent pregnancy loss (RPL). The subject presented herein is written in a descriptive manner and not as a systematic review so that the rationale for the use of heparins and the consequences of their biological activity can be brought to readers outside of our profession.

The Physiological Role of Heparin

Heparin is one of the oldest drugs currently in widespread clinical use. Its discovery in 1916 predates the establishment of the Food and Drug Administration of the United States, although it did not enter clinical trials until 1935. It was originally isolated from canine liver cells (Linhardt, 1991), hence its name (hepar or “ήπαρ” is Greek for “liver”).

It is principally employed for its anticoagulation properties. Moreover, its true physiological role in the body remains uncertain, since blood anticoagulation is achieved mostly by heparan sulfate proteoglycans derived from endothelial cells (Marcum et al., 1986). Heparin is usually stored within the mast cells secretory granules and released only into the vasculature at sites of tissue injury. It has been proposed that, in addition to its anticoagulant properties, heparin may play a role in the defense against invading bacteria and other foreign materials (Nader et al., 1999).

Heparin is a glycosaminoglycan composed of chains of alternating residues of d-glucosamine and uronic acid. Its major anticoagulant effect is accounted for in a unique pentasaccharide (GlcNAc/NS(6S)-GlcA-GlcNS(3S,6S)-IdoA(2S)-GlcNS(6S) structure that has a high binding affinity sequence to anti-thrombin III (AT-III) (Rosenberg & Bauer, 1992); however, in-vitro studies suggest that this structure is present only in about one third of heparin molecules (Rosenberg et al., 1979).

The interaction between heparin and AT-III mediates the majority of the anticoagulant effect of the former. Their binding produces a conformational change in AT-III (Fig. 1) that accelerates up to 1,000 fold (Björk & Lindahl, 1982) its ability to inactivate the major coagulation factors, including mainly thrombin (factor IIa), factor Xa, and factor IXa (Rosenberg & Bauer, 1992).

Heparin increases the inhibitory effect of AT-III on thrombin and Factor Xa activity by distinct mechanisms (Fig. 2). The acceleration of the inhibition of thrombin by AT-III necessitates the binding of this molecule to the heparin polymer proximally to the pentasaccharide units. Heparin has a highly negative charge that is derived from the number of its saccharide units, which contributes to the strong electrostatic interaction of AT-III with thrombin. Thus, heparin’s activity against thrombin is size-dependent, and the ternary complex (including thrombin, AT-III and heparin) requires at least 18 saccharide units for efficient formation and thrombin inactivation (Petitou et al., 1987; Petitou et al., 1997).

In contrast, the effect of heparin on the inhibition of factor Xa by AT-III is dependent on the conformational change of this molecule at the heparin-binding site; therefore, the size of heparin has no importance in the inhibition of factor Xa by AT-III. This effect has therapeutic implications and led to the development of a new generation of heparin derived anticoagulants including low molecular weight heparins (LMWH) and fondaparinux. LMWH are obtained as fragments of unfractionated heparin as a result of enzymatic or chemical depolymerization, yielding to molecules of mean weight of 5,000 Da (Table 1) (Weitz, 1997) while fondaparinux is a synthetic pentasaccharide based on the heparin antithrombin-binding domain (Chang et al., 2014).

Figure 1 Anti-thrombin III after conformational change induced by heparin binding.

Reproduced with permission from Whisstock et al. (2000).

Figure 2 Mechanisms of interaction between heparin, anti-thrombin III, thrombin (A) and factor Xa (B).

Table 1 Comparison among low molecular weight heparin preparation.

Reproduced with permission from Weitz (1997).

Preparation	Method of
preparation	Mean molecular
weight	Anti-Xa:anti-IIa
ratioa	
Ardeparin (Normiflo)	Peroxidative depolymerization	6,000	1.9	
Dalteparin (Fragmin)	Nitrous acid depolymerization	6,000	2.7	
Enoxaparin (Lovenox)	Benzylation and alkaline depolymerization	4,200	3.8	
Nadroparin (Fraxiparine)	Nitrous acid depolymerization	4,500	3.6	
Reviparin (Clivarine)	Nitrous acid depolymerization, chromatographic
purification	4,000	3.5	
Tinzaparin (Innohep)	Heparinase digestion	4,500	1.9	
Notes.

a The ratios were calculated by dividing the anti–factor Xa (anti-Xa) activity by the antithrombin (anti-IIa) activity. The ratios are based on information provided by the manufacturers.

These medications target the anti-factor Xa activity rather than anti-thrombin (IIa) activity of AT-III, aiming to facilitate a more subtle regulation of coagulation with an improved therapeutic index and less side effects. Indeed, each molecule of fondaparinux binds to one molecule of AT-III at a specific site, and with very high affinity. The binding is rapid, non-covalent, and reversible. It induces a critical conformational change in AT-III, exposing a loop containing an arginine residue that binds factor Xa. Exposure of the arginine-containing loop greatly increases the affinity of AT-III for factor Xa, potentiating the natural inhibitory effect of AT-III against factor Xa by a factor of approximately 300 (Petitou et al., 1987; Petitou et al., 1997).

The Role of Heparins in Implantation and Placentation

What are the stages of implantation and placentation?

Implantation, a critical step for the establishment of pregnancy, requires complex molecular and cellular events resulting in uterine growth and differentiation, blastocyst adhesion, invasion, and placental formation. Successful implantation necessitates a receptive endometrium, a normal and functional embryo at the blastocyst stage, and a synchronized dialogue between the mother and the developing embryo (Dey et al., 2004). In addition to the well-characterized role of sex steroids, the complexity of blastocyst implantation and placentation is exemplified by the role played by a number of cytokines and growth factors in these processes. Indeed, the process of implantation is orchestrated by hormones like sex steroids and hCG; growth factors such as TGF-B, HB-EGF, IGF-1; cytokines as Leukemia Inhibitory Factor, Interleukin-6 and Interleukin-11; adhesion molecules including L-selectin and E-cadherin; the extracellular matrix (ECM) proteins; and prostaglandins (Dey et al., 2004).

Embryonic implantation is initiated by the recognition and adhesion between the blastocyst surface and the uterine endometrial epithelium. Adhesion occurs when a free-floating blastocyst comes into contact with the endometrium during the ‘receptive window’ in which it is able to respond to the signals from the blastocyst. This contact is then stabilized in a process known as adhesion in which the trophoblast cells establish contact with the micro protrusions present on the surface of the endometrium known as pinopodes (Lopata, Bentin-Ley & Enders, 2002). The last step of implantation is the invasion process, which involves penetration of the embryo through the luminal epithelium into the endometrial stroma; this activity is mainly controlled by the trophoblast.

The trophoblast lineage is the first to differentiate during human development, at the transition between morula and blastocyst. Initially, at day 6–7 post-conception, a single layer of mononucleated trophoblast cells surrounds the blastocoel and the inner cell mass. At the site of attachment and direct contact to maternal tissues, trophoblast cells fuse to form a second layer of postmitotic multinucleated syncytiotrophoblast (Hoozemans et al., 2004). Once formed, the syncytiotrophoblast grows by means of steady incorporation of new mononucleated trophoblast cells from a proximal subset of stem cells located at the cytotrophoblast layer (Jauniaux, 2000).

Tongues of syncitiotrophoblast cells begin to penetrate the endometrial cells, and gradually the embryo is embedded into the stratum compactum of the endometrium. A plug of fibrin initially seals the defect in the uterine surface, but by days 10–12 the epithelium is restored (Hertig, Rock & Adams, 1956). Only at around the 14th day do mononucleated cytotrophoblasts break through the syncytiotrophoblast layer and begin to invade the uterine stroma at sites called trophoblastic cell columns. Such cells constitute the extravillous trophoblast, and have at least two main subpopulations: interstitial trophoblast, comprising all those extravillous trophoblast cells that invade uterine tissues and that are not located inside vessel walls and lumina; and endovascular trophoblast, located inside the media or lining the spiral artery lumina and partly occluding them (sometimes this subtype is further subdivided into intramural and endovascular trophoblast) (Hertig, Rock & Adams, 1956).

At a molecular level, trophoblast adhesion from the stage of implantation onwards is an integrin-dependent process (Damsky, Fitzgerald & Fisher, 1992; Zhou et al., 1997) that takes place in a chemokine- and cytokine- rich microenvironment analogous to the blood-vascular interface. Of note, uterine expression of chemokines in humans is hormonally regulated and the blastocyst expresses chemokine receptors. In addition, oxygen tension plays an important role in guiding the differentiation process that leads to cytotrophoblast invasion to the uterus (Lash et al., 2006; Zhao et al., 2012).

What is the role of heparin and heparin derived molecules in the process of implantation?

Heparin and heparin derived molecules influence all stages of implantation. This anticoagulant has an effect on the expression of adhesion molecules, matrix degrading enzymes and trophoblast phenotype and apoptosis (see Table 2).

Table 2 Overview of molecules involved in the process of implantation, trophoblast development and placentation, and effect of heparin on these molecules.

Molecule	Site of expression	Activity	Effect of heparin	
Anti-thrombin III	Maternal circulation Trophoblast	Inactivation of coagulation factors, including mainly thrombin (factor IIa), factor Xa, and factor IXa	Conformational change in AT-III that accelerates its ability to inactivate the coagulation factors	
Selectins (E- P- and L-selectins)	E-selectin endothelium, P-selectin platelets, L-selectin leucocytes and blastocyst surface	Cell adhesion and homing	Interference with inflammatory cells adhesion and homing but probable interference with blastocyst decidual adhesion	
Cadherins	Trophoblast, placenta, decidua	Cell adhesion (invasive phenotype acquired in case of reduction of expression)	Reduction of expression	
Heparin-binding EGF-like growth factor (HB-EGF)	Trophoblast and placenta	(1) Potent mitogen and chemoattractant in its soluble form (2) Promotion of adhesion of the blastocyst to the uterine wall in a mouse-in-vitro-system (3) Regulation of the conversion of human cytotophoblast into invasive phenotype and influence on the motility of these cells (4) Prevention of hypoxic induced apoptosis	Increased decidual expression and secretion of HB-EGF	
Matrix metalloproteinases (MMPs)	Soluble form	Involvement in trophoblast invasion into endometrial tissue	Increased expression	
Tissue inhibitors of metalloproteinases (TIMPs)	Soluble form	Inhibition of metalloproteinases and their function	Reduction of expression	
Macrophage antigen 1 (Mac1)	Surface of myeloid cells	Coordination of adhesive functions of leukocyte and their migration	Interference with myeloid cell adhesion and transmigration	
Platelet/endothelial cell adhesion molecule 1 (PECAM1)	Surface of platelets, endothelia, monocytes, neutrophils, T-cell subsets and granulocyte/macrophage precursors	Transmigration of inflammatory cells through the endothelial wall	Interference with inflammatory cells transmigration	

Selectins and cadherins

Selectins and cadherins families are the main adhesion molecules investigated with regard to the implantation process. Selectins are a group of three carbohydrate-binding proteins that are named following the cell type expressing them (E- endothelium, P- platelets, and L- leucocytes): E-selectin is expressed on the endothelial surface; P-selectin on the surface of activated platelets; and L-selectin on lymphocytes, where it plays an essential role in the homing mechanism of these cells (Rosen, 2004; Rosen, 2006). The selectins adhesion system may constitute an initial step in the implantation process. Indeed, L-selectin is strongly expressed on the blastocyst surface while, during the window of implantation, there is an up-regulation in the decidual expression of the selectin oligosaccharide-based ligands, predominantly on endometrial luminal epithelium (Genbacev et al., 2003). This may assist in the blastocyst decidual apposition during the implantation process.

The effect of heparin on selectins during implantation is unclear. Due to its high density in negatively charged sulfates and carboxylates, heparin is able to bind the two binding sites of the natural ligand of selectin molecules (P and L-selectins: one for the sialyl Lewis X moiety and another for the tyrosine sulfate-rich region of its native ligand P-selectin glycoprotein ligand-1 [PSGL-1]), and the number of sites bonded is dependent on the length of the heparin chain. Evidence in support is presented by the study of Stevenson, Choi & Varki (2005) who investigated the effect of different unfractionated heparin and LMWH on selectin molecules in cancer cell lines. Tinzaparin, with 22% to 36% of fragments greater than 8 kDA, significantly impaired L-selectin binding to its ligand; whereas enoxaparin, with 0% to 18% fragments greater than 8 kDa, did not affect L-selectin expression (Stevenson, Choi & Varki, 2005). Thus, heparins with high proportion of fragments longer than 8 kDa may reduce inflammatory cell adhesion and homing; on the other hand, they may affect blastocyst adhesion by blocking selectins ligand binding sites.

Cadherins are a group of cell adhesion proteins that mediate Ca2+-dependent cell–cell adhesion, a fundamental process required for blastocyst implantation and embryonic development (Frenette & Wagner, 1996). E-cadherin plays an important role in maintaining cell adhesion. In cancer cells, the reduction of E-cadherin expression promotes acquisition of invasive phenotype. Interestingly, gestational trophoblastic diseases (choriocarcinoma and complete hydatidiform mole), that are characterized by invasive trophoblast behavior, has a lower E-cadherin trophoblastic expression than that of first-trimester placenta (Xue et al., 2003). In contrast, the trophoblast expression of E-cadherin is higher in placentas of patients with preeclampsia than in those of normal pregnant women (Li et al., 2003). The effect of heparin on E-cadherin expression was studied by Erden et al. (2006), who randomly treated female rats with different heparins (UFH, enoxaparin, and tinzaparin) during the preconceptional period, and examined E-cadherin expression in tissue sections of placenta and decidua from the different groups. The group treated by UFH had a lower E-cadherin placental staining than other study groups. In addition, the decidual staining score of this molecule was lower both in the UFH and Enoxaparin groups in comparison to controls and rats treated with tinzaparin. Therefore, there is evidence to support the effect of heparins on trophoblast invasiveness through E-cadherin expression, providing a possible mechanism by which heparin could promote trophoblast cell differentiation and motility.

Heparin binding EGF-like growth factor

Heparin-binding EGF-like growth factor (HB-EGF) is a 76–86 amino acid glycosylated protein that was originally cloned from macrophage-like U937 cells. It is a member of the epidermal growth factor (EGF) family that stimulates growth and differentiation. HB-EGF utilizes various molecules as its “receptors”. The primary receptors are in the ErbB (also named HER) system, especially ErbB1 and ErbB4, human tyrosine kinase receptors. HB-EGF is initially synthesized as a transmembrane precursor protein, similar to other members of the EGF family of growth factors. The membrane-anchored form of HB-EGF (pro HB-EGF) is composed of a pro domain followed by heparin-binding, EGF-like, juxtamembrane, transmembrane and cytoplasmic domains. Subsequently, proHB-EGF is cleaved at the cell surface by a protease to yield the soluble form of HB-EGF (sHB-EGF) using a mechanism known as ectodomain shedding. sHB-EGF is a potent mitogen and chemoattractant for a number of different cell types. Studies of mice expressing non-cleavable HB-EGF have indicated that the major functions of HB-EGF are mediated by the soluble form (Miyamoto et al., 2006).

HB-EGF accumulates in the trophoblast (Cha, Sun & Dey, 2012) throughout the placenta (Leach et al., 1999). Multiple roles for this growth factor are suggested by its cell specific expression during the human endometrial cycle and early placentation, and high levels expression in the first trimester (Yoo, Barlow & Mardon, 1997).

The membrane active precursor functions as a justacrine growth factor and cell-surface receptor. It has been demonstrated to promote adhesion of the blastocyst to the uterine wall in a mouse-in-vitro-system (Raab & Klagsbrun, 1997) suggesting a role for HB-EGF in embryo attachment to the uterine luminal epithelium. As stated above, the majority of its biological functions are mediated by its mature soluble form. A major role in early stages of placentation is represented by cellular differentiation and consequent invasion of the uterine wall and vascular network.

Several changes occur in the expression of adhesion molecules as cytotrophoblast differentiation proceeds, which results in pseudovasculogenesis or the adaptation by cytotrophoblast to a molecular phenotype that mimics endothelium (Zhou et al., 1997). For example, during extravillous differentiation in vivo, integrin expression is altered from predominantly α6β4 in the villous trophoblast to α1β1 in cytotrophoblasts migrating throughout the decidual stroma (Damsky, Fitzgerald & Fisher, 1992) or engaging in endovascular invasion (Zhou et al., 1997).

Leach et al. (2004) demonstrated the role of HB-EGF in regulating the conversion of human cytotrophoblasts into invasive phenotype and the motility of these cells. This study demonstrated the ability of HB-EGF to induce ‘integrin switching’ through intracellular signaling following ligation of HER tyrosine kinases, altering integrin gene expression to stimulate cytotrophoblast invasion at a molecular level.

In addition to its effect on the invasive trophoblast phenotype, HB-EGF can affect cell motility. Indeed, cytotrophoblast motility was specifically increased by each of the EGF family members examined. The expression by cytotrophoblasts of each growth factor, as well as their receptors, suggests the possibility of an autocrine loop that advances cytotrophoblast differentiation to the extravillous phenotype.

The ability of the HB-EGF molecule to prevent hypoxic induced apoptosis plays a fundamental role in early stages of placentation. During the entire 1st trimester, the organogenesis period, embryonic development takes place in a low O2 tension environment. Oxygen concentration is relatively low (18 mmHg or 2%) at the human implantation site through the first 10 weeks of gestation due to occlusion of the uterine spiral arteries by extravillous trophoblasts. Oxygen availability serves as a developmental cue to regulate trophoblast proliferation. Experimental evidence suggests that this environment is essential for both fetal and placental development, and premature exposure to normal oxygen concentrations is associated with increased rate of pregnancy complications such as preeclampsia, IUGR and miscarriage (Jauniaux et al., 2003).

First trimester human cytotrophoblast cell survival at 2% O2 is dependent on HB-EGF signaling (Armant et al., 2006). Indeed, HB-EGF expression is up regulated by hypoxia, and it functions as a mitogen and potent cell survivor factor during stress. The mechanism proposed for this effect of HB-EGF is as follows: sHB-EGF is released by activated metalloproteinases that cleave the extracellular domain of proHB-EGF. sHB-EGF binds to HER1 or HER4 through its EGF-like domain and to heparan sulfate proteoglicans (HSPG) through its heparin binding domain, and this is followed by receptor homo- or heterodimerization with other members of the HER family. Subsequent transphosphorylation of HER cytoplasmatic domains at tyrosine residues initiates a downstream signaling that increases proHB-EGF accumulation and inhibits apoptosis. This positive feedback loop upregulates HB-EGF secretion to achieve extracellular HB-EGF levels sufficient to maintain cell survival at 2% O2 (Armant et al., 2006).

As a result, HB-EGF has a fundamental role in successful pregnancies. This molecule mediates a vast number of functions beginning from the earliest stages of pregnancy and up to term, ranging from adhesion, to implantation and invasion, successful placentation, and protection from hypoxic induced aptoptosis. The effect of heparin on this molecule is currently being studied. Di Simone et al. (2012) demonstrated that LMWH induced an increased decidual expression and secretion of HB-EGF in a dose-dependent manner. A different study by D’Ippolito et al. (2012) demonstrated that LMWH induces activation of Activator Protein-1 (AP-1), a DNA-binding transcription factor which regulates the expression of HB-EGF. Activated AP-1 translocates to the nucleus and binds the promoter region of HB-EGF gene, thus enhancing its protein expression. Hills et al. (2006) demonstrated that heparin is capable of activating the EGF receptor in primary villous trophoblast.

Thus, we propose that accumulating evidence suggests that the beneficial effect of heparin in preventing placental mediated pregnancy complications may derive from its effect on HB-EGF expression and concentration, especially during the first trimester.

Matrix metalloproteinases

In addition to the adhesion molecules, matrix metalloproteinases (MMPs) are an important component in the process of blastocyst implantation. MMPs are a group of matrix degrading enzymes which are secreted as inactive zymogen and must be cleaved to become active (Isaka et al., 2003). Among the members of the MMP family, MMP-2 and MMP-9 type IV collagenases were suggested to be involved in trophoblast invasion into endometrial tissue (Librach et al., 1991). Indeed, the profile of proMMP 2 and 9 secretion differs during the stages of trophoblast invasion and implantation, and differences in these zymogens expression were found between 6–8 and 9–12 weeks of gestation in extravillous cytotrophoblast cells (Staun-Ram et al., 2004). Di Simone et al. (2007) investigated the effect of LWMH specifically on placental MMPs, and the degrading capacity of the trophoblast cells. This effect is mediated by the action of heparins on both metalloproteinases (MMPs) and their tissue inhibitors (TIMPs). Heparin increased both the MMPs concentration and activity by affecting their transcription, the conversion of the proenzyme into the active form, and the reduction of the synthesis of the specific inhibitors TIMPs (both the mRNA and protein levels) in a dose dependent manner (Di Simone et al., 2007).

Immunologic and anti-inflammatory effects of heparins

Immune tolerance of the semi-allogeneic fetus is the prerequisite for a successful pregnancy outcome (Clark, Arck & Chaouat, 1999). The maternal blood is in direct contact with the syncytiotrophoblast at the intervillous space. In addition, the extravillous trophoblast that anchors the placenta to the decidua, and further differentiate into endovascular trophoblast that invades spiral arteries and remodels the vessel walls, is also in direct contact with the maternal blood (Norwitz, Schust & Fisher, 2001; Bischof & Campana, 1996; Red-Horse et al., 2004). Both innate and adaptive immune responses contribute to a maternal fetal cross-talk that balances the anti- and pro-inflammatory processes in the feto-maternal interface (Norwitz, Schust & Fisher, 2001; Moffett-King, 2002). These processes involve: MHC class I molecules, hormones, complement regulatory proteins, immunoregulatory molecules (i.e., indolamine 2,3-dioxygenase, Fas/Fas- Ligand, IL-10), regulatory T cells (CD4+ CD25+ Foxp3+), regulatory macrophages, and growth factors expressed at the placental–decidual interface (Redecha et al., 2007; Thangaratinam et al., 2011; Wegmann et al., 1993; Thellin et al., 2000; Li & Huang, 2009; Mjösberg et al., 2007; Karimi, Blois & Arck, 2008; Aluvihare, Kallikourdis & Betz, 2004). These mechanisms act in concert to sustain the maternal tolerance to the semi-allogenic placenta and fetus (Kalkunte et al., 2009). In addition to its well-understood anticoagulant activity, heparin also has an impact on the immune system (Martz & Benacerraf, 1973; Sy et al., 1983; Arfors & Ley, 1993). The main known effect of heparin is on the migration and adhesion of leukocytes during an inflammatory response (Stevenson, Choi & Varki, 2005).

The anti-inflammatory effects of heparin are derived from several mechanisms: (1) the molecular structure of heparin is so that, upon its binding to the endothelial cells of blood vessels, it creates a negatively charged surface that is facing the vessel lumen. These negatively charged molecules repulse the negatively charged leukocytes and prevent their adhesion to the endothelium (heparan sulfate molecules that are expressed on leukocyte surfaces are responsible for the negative charge of these cells); (2) heparin is a large molecule that can bind a substantial number of proteins which play an important role in inflammation including selectins (L-selectin (Koenig et al., 1998) and P-selectin molecules (Skinner et al., 1991)) and integrins. The B2-integrin adhesion molecule CD11b/CD18, also known as Macrophage antigen 1 (Mac1), is a member of a subfamily of related cell-surface glycoproteins that coordinate adhesive functions including leukocyte migration (Kishimoto et al., 1989). Mac1 is expressed on myeloid cells and binds to molecules as intercellular adhesion molecule 1 (ICAM1), fibrinogen, iC3b, and factor Xa. The heparin-Mac1 bond interferes with myeloid cell adhesion and transmigration (Diamond et al., 1995). Heparin also binds to platelet/endothelial cell adhesion molecule 1 (PECAM1), a member of the Ig superfamily, expressed on a variety of cells such as platelets, endothelia, monocytes, neutrophils, T-cell subsets and granulocyte/macrophage precursors. This molecule is involved in homotypic and heterotypic cellular adhesion and plays a role in the transmigration of inflammatory cells through the endothelial wall. Heparin is capable of binding PECAM1 and interfering with its action (Watt et al., 1993), thus reducing the effectiveness of the inflammatory response.

The anti-inflammatory properties of LMWH have been demonstrated within in vivo models. Indeed, Wang et al. (2013) investigated the effects of LMWH on dextran sulfate sodium (DSS)-induced colitis in a mice model. The authors reported that mice which were treated with LMWH had a significant decrease in the expression of both IL-1β and of IL-10 mRNA, leading to a down regulation of inflammatory cytokines production. LMWH also imitates the function of Syndecan-1 (a protein which expression is inversely correlated to the mRNA expression of IL-1β in the intestinal mucosa of DSS-induced colitis models). It plays an important role in promoting wound repair, maintaining cell morphogenesis, and mediating inflammatory responses (Götte, 2003) by aiding the clearance of pro-inflammatory chemokines. In addition Li et al. (2013) found that treatment with UFH can attenuate inflammatory responses of lypopolisaccharide induced acute lung injury in rats. The mechanisms by which UFH exerts its anti-inflammatory effect seem to correlate with its inhibition on IL-1ß and IL-6 production via inactivation of the NF-κB pathways.

In humans the anti-inflammatory activity of heparin has been evidenced by small clinical trials in patients suffering from a range of inflammatory diseases (Gaffney & Gaffney, 1996), including rheumatoid arthritis and bronchial asthma. Remission of disease has been described in nine out of ten patients with refractory ulcerative colitis treated with combined heparin and sulphasalazine (Gaffney & Gaffney, 1996). A subjective improvement of asthma symptoms using intravenous heparin is described (Fine, Shim & Williams, 1968; Boyle, Smart & Shirey, 1964), while other studies with inhaled heparin demonstrated reduced bronchoconstrictive responses in patients with exercise-induced asthma (Garrigo, Danta & Ahmed, 1996; Ahmed et al., 1993).

The clinical rationale for the use of heparin in the treatment of inflammatory diseases may be based on the fact that many of the molecular mechanisms involved in tumor metastasis are the same responsible for cell recruitment in inflammation; heparin has been successful in treating both conditions (Tyrrell et al., 1999).

Clinical Application of the use of Heparins During the First Half of Pregnancy

In light of the possible effects of heparins and heparin binding molecules on the blastocyst implantation and placentation, this family of drugs may play a role in the prevention of RIF in IVF patients and in the treatment of patients with RPL.

Is there a benefit of the use of heparins in the prevention of recurrent implantation failure in IVF patients?

The term RIF has been used since 1983 to describe the failure of embryos to implant following IVF treatments. There is no unanimous definition for RIF in terms of the number of failed cycles or the total number of transferred embryos that have not successfully implanted. The ESHRE PGD consortium document (Thornhill et al., 2005) mentioned that RIF can be considered after more than three high-quality embryo transfers or implantation failure with transfer of ≥10 embryos in multiple transfers with exact numbers to be determined by each center. In order to improve pregnancy outcomes in women with RIF, various investigations and treatment adjuncts including heparin have been studied.

A recent meta-analysis with systematic review of the literature (Potdar et al., 2013) included randomized controlled, quasi-randomized and prospective studies comparing the use of LMWH with placebo or no adjuvant treatment in women with RIF undergoing IVF/ICSI. After the process of literature search and selection, one quasi-randomized (Potdar et al., 2013) and two randomized (Amarin et al., 2008; Urman et al., 2009) studies were selected for the meta-analysis and included 243 women with RIF who underwent IVF/ICSI, 127 in the intervention group and 116 in the control/placebo.

All three studies were unclear for detection bias, and none of the studies explicitly stated whether the individuals assessing the outcome were blinded to the trial or not. However, assessment for pregnancy outcome is unlikely to be subjective since implantation, clinical pregnancy, multiple pregnancies and miscarriage are all objectively assessed during the ultrasound scan.

The results of this meta-analysis show that, in women with ≥3 RIF, the use of LMW as an adjunct to IVF treatment significantly improved the life birth rate by 79%.

This result suggests that there could be a potential role of LMWH in improving pregnancy outcomes for women with RIF.

What is the effect of heparins on pregnancy success in women with recurrent pregnancy loss?

A beneficial effect of antithrombotic agents (heparin in particular) in women with RPL was already suggested in 1980 (Langer et al., 1980). Different clinical definitions of this condition were arrived at by the different scientific societies: the Royal College of Obstetricians and Gynecologists (RCOG) considers three or more first trimester miscarriages as RPL, while the American Society for Reproductive Medicine (ASRM) establishes a limit of two or more pregnancy losses.

Many underlying mechanisms have been recognized for RPL, including chromosomal defects (Rajcan-Separovic et al., 2010), endocrinopathies (Ke, 2014) (thyroid diseases and diabetes), uterine malformations (Jaslow, 2014), and autoimmune diseases (Kwak-Kim et al., 2013). In addition, thrombophilic mutations (Lino et al., 2014) have been suggested as leading to alterations in embryonic formation, migration, implantation and placentation; in the past three decades, this area has become a field of extensive study with the goal of increasing the rate of live births in these patients.

Conflicting results are provided from the studies performed to address this question. This is also a consequence of the marked heterogeneity among the different studies. Evidence in support of this view is shown by several experiences, including prospective (Brenner et al., 2000) and retrospective (Carp, Dolitzky & Inbal, 2003) cohort studies, which evaluated the benefit of treatment with heparin in patients with RPL in terms of live birth rate. Moreover, due to several criticisms contained in the study design, there is a need for randomized clinical trials to further address the question of whether heparins could be beneficial in patients with RPL by comparing the outcomes of live birth rate among treated and untreated women. The following RCTs were performed for this purpose, including both patients with or without inherited thrombophilia: (1) In the LIVE-ENOX study (Brenner et al., 2005), 180 women with thrombophilia and RPL were randomized to either enoxaparin 40 or 80 mg once daily. In addition to inherited thrombophilia, women with antiphospholipid antibodies, MTHFR 677TT genotype and hyperhomocysteinemia were eligible. The live birth rates in both groups were similar (84.3% and 78.3%, respectively), but as a control group was lacking, the effect of enoxaparin could not be validated; (2) In 2006, Dolitzky et al. (2006) randomized 54 patients with RPL either for treatment with enoxaparin or aspirin, considering subsequent live births or miscarriage as the main outcome. Both groups had a similar live birth rate (RR 0.92, 95% CI [0.58–1.46]). This study showed that, even though there were not statistical difference between the study groups, women who were treated by heparins had a higher live birth rate to that reported in literature on women with RPL; (3) The ALIFE study (Kaandorp et al., 2010) included 364 women with two or more unexplained pregnancy losses that were randomized to nadroparin 2,850 International Unit combined with aspirin 80 mg, aspirin 80 mg only, or a placebo before conception or at a maximum gestational age of 6 weeks. Of these women, 299 became pregnant. The chance of live birth did not differ among the treatment groups. The relative risk of live birth for women who became pregnant was 1.03 (95% CI [0.85–1.25]) for nadroparin combined with aspirin, and 0.92 (95% CI [0.75–1.13]) for aspirin only compared with placebo. The study was not designed to evaluate the beneficial effect of heparin on thrombophilic patients, but the author performed a secondary analysis on the effect of heparins according to the presence of inherited thrombophilia, which showed no significant difference in the primary outcome among the groups; (4) A randomized double-blind (for aspirin) multicenter trial (Visser et al., 2011) was performed among 207 women with three or more consecutive first trimester (<13 weeks) miscarriages, two or more second trimester (13–24 weeks) miscarriages, or one third trimester fetal loss combined with one first trimester miscarriage. Women underwent workup for thrombophilia and were randomly allocated before seven weeks gestation to either enoxaparin 40 mg + placebo (n = 68), enoxaparin 40 mg + aspirin 100 mg (n = 63) or aspirin 100 mg (n = 76). The primary outcome was the live-birth rate. The trial was stopped prematurely because of slow recruitment. A live birth rate of 71% (RR 1.17, 95% CI [0.92–1.48]) was found for enoxaparin and placebo and 65% (RR 1.17, 95% CI [0.92–1.39]) for enoxaparin and aspirin when compared to aspirin alone (61%, reference group). In the whole study group, the live birth rate was 65% (95% CI [58.66–71.74]) for women with three or more miscarriages (n = 204). No difference in pregnancy complications, neonatal outcome or adverse effects was observed. No significant difference in live birth rate was found with enoxaparin treatment versus aspirin or a combination of both versus aspirin in women with recurrent miscarriage.

Little evidence is available for the effect of antithrombotic agents in women with a single pregnancy loss and inherited thrombophilia. The results from several small retrospective and prospective cohort studies in women with inherited thrombophilia, with or without previous pregnancy complications, suggest a beneficial effect of antithrombotic therapy to reduce pregnancy complications (Kupferminc et al., 2001; Kupferminc et al., 2007). These studies are heterogeneous with regard to study design and study population.

Conclusion

Heparins play a role in embryonic implantation and placentation, and contribute to the development of a normal pregnancy. This effect is gained through the interaction of heparins with coagulation factors, anticoagulation proteins, their effect on the expression of adhesion molecules, matrix degrading enzymes and trophoblast phenotype and apoptosis: all important components in the process of embryonic implantation and placentation.

The fact that heparins may play a role in implantation and placentation led to their use in the prevention of RIF and RPL. In RIF heparins demonstrated a beneficial effect that could be attributed to the effects of this molecule on enhancing endometrial receptivity and trophoblast invasion due to the regulation of heparin-binding factors, adhesion molecules or inhibition of complement activation. In contrast, the positive effect of heparin as a treatment for RPL is not that clear. One possible explanation for this lack of conclusive evidence is the syndromic nature of RPL which results from several underlying mechanisms of disease. Thus, heparins may have a role in improving pregnancy outcomes among a subset of patients with RPL regardless to the presence of thrombophilia, but a conclusive statement in this matter awaits further investigation.

Additional Information and Declarations

Competing Interests

Author Contributions

Offer Erez is an Academic Editor for PeerJ.

Michela Quaranta, Offer Erez, Salvatore Andrea Mastrolia, Arie Koifman, Elad Leron, Tamar Eshkoli, Moshe Mazor and Gershon Holcberg wrote the paper, prepared figures and/or tables, reviewed drafts of the paper.

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
