# Peer review of "The physiologic and therapeutic role of heparin in implantation and placentation"

_PeerJ, doi:10.7717/peerj.691_

## Round 0.1 · original submission · Major Revisions

· Academic Editor

Major Revisions

The reviewers found merit in your paper but also lack of practical information that preclude publishing it in its present form. The manuscript will be improved by addition of information on clinical data obtained by randomized controlled or observational studies and indications about the use of heparin in clinical practice during pregnancy. Practical implications of the in vitro findings should be discussed.

Reviewer 1 ·

Basic reporting

The manuscript is in a well written English

Experimental design

no comments

Validity of the findings

Conclusions are adequate.

Additional comments

The manuscript is well written. Authors should add a final paragraph reporting
- indications to use of heparins in clinical practice during pregnancy
- practical implications of the reported in vitro findings, in terms of future research or future clinical trials.

Reviewer 2 ·

Basic reporting

The authors report on mechanisms possibly involved in the improvement of implantation and placentation by heparins.

Experimental design

This is not a systematic review, it seems a "narrative" review, but this is not clearly reported by authors. If I have correctly understood the Scope of the journal this type of article is not acceptable.

Validity of the findings

This review focuses on the role played by heparins in implantation and placentation.
Data are properly reported and the paper is well written. However, the review could be improved by adding information on clinical data from RCTs or observational studies.

Additional comments

This review is interesting and well written. It could be improved by adding clinical data from RCTs or observational studies.

---

## Round 0.2 · accepted · Accept

· Academic Editor

Accept

The revised version of the manuscript is improved and it's now acceptable for publication.